# Utility of Measuring Circulating Bio-Adrenomedullin and Proenkephalin for 30-Day Mortality Risk Prediction in Patients with COVID-19 and Non-COVID-19 Interstitial Pneumonia in the Emergency Department

**DOI:** 10.3390/medicina58121852

**Published:** 2022-12-15

**Authors:** Ilaria Dafne Papasidero, Gabriele Valli, Dario Marin, Alberto Del Sasso, Antonio De Magistris, Elisa Cennamo, Silvia Casalboni, Francesca De Marco, Roberta Rocchi, Brice Ndogmo Beumo, Valeria Cusani, Mariarosa Gaudio, Oliver Hartmann, Andreas Bergman, Maria Pia Ruggieri, Salvatore Di Somma

**Affiliations:** 1Postgraduate School of Emergency Medicine, Sapienza University of Rome, 00185 Rome, Italy; 2Department of Emergency Medicine, San Giovanni Addolorata Hospital, 00184 Rome, Italy; 3Department of Clinical Pathology, San Giovanni Addolorata Hospital, 00184 Rome, Italy; 4SphingoTec, 16761 Hennigsdorf, Germany; 5Global Research on Acute Conditions Team (Great Network), 00191 Rome, Italy; 6Department of Medical-Surgery Sciences and Translational Medicine, University of Rome Sapienza, 00185 Rome, Italy

**Keywords:** COVID-19, interstitial pneumonia, Bio-ADM, penKid, acute kidney injury, 30-day mortality

## Abstract

*Background and Objectives*: In order to accelerate the risk stratification of patients referred to the Emergency Department (ED) with interstitial pneumonia, it could be useful to provide new and effective laboratory tests for use. The aim of our study was to evaluate the prognostic role of two biomarkers, bio-adrenomedullin (Bio-ADM) and proenkephalin (penKid), in patients with interstitial pneumonia (IP) at ED admission. *Materials and Methods*: In 153 consecutive patients with IP, both from COVID-19 or non-COVID-19 etiology, we measured, in a prospective observational manner, penKid and Bio-ADM at ED admission and after 24 h. In order to evaluate patient outcomes, 30-day follow-ups were also performed. The endpoints were 24 h, 10-day, and 30-day mortality. *Results*: Both biomarkers were shown to be good predictors of adverse events at 30 days, with Bio-ADM outperforming penKid. Bio-ADM was linked with 24 h and 10-day patient mortality. Moreover, PenKid was related to parameters defining worsening kidney function. *Conclusions:* Both in patients with COVID-19 or non-COVID-19 interstitial pneumonia at ED admission, Bio-ADM and penKid were good predictors of patient mortality. To evaluate these two biomarkers could be considered to be useful during the first evaluation in the ED when integrated with clinical scores.

## 1. Introduction

Severe acute respiratory syndrome coronavirus-2 (SARS-CoV-2) is the novel coronavirus that causes coronavirus disease 2019 (COVID-19). The first cases of COVID-19 were reported in the Chinese city of Wuhan located in the province of Hubei in December 2019 [1]. Since the initial detection of the virus, more than 554 million cases of COVID-19 have been confirmed as of 18th of November 2022 with more than 6.5 million deaths [2]. The epidemiological and clinical characteristics of patients with COVID-19 have been widely reported, but its pathogenesis, risk factors for mortality, and detailed clinical course of illness have not been fully described [3].

COVID-19-infected cases present with symptoms such as fever, dry cough, and other flu-like findings, but with a possible rapid deterioration with lower airway involvement and complicated by pneumonia [3,4]. In these subjects, the disease evolves into severe interstitial pneumonia with acute impairment of gas exchange and sudden clinical worsening within 5–10 days of infection [5]. The factors leading to the sudden clinical deterioration of these patients have not been clearly identified with the exception of generic risk factors such as age, sex, and comorbidities that typically identify frail patients [6]. Patients affected by this syndrome may need extensive monitoring and progressively more complex treatments according to the severity of the disease [7]. Furthermore, in the presence of Emergency Department (ED) overcrowding, these patients could represent a matter for final disposition and could wait in the emergency room for a long time [8], also with a significant increase in related costs [9]. Consequently, for these subjects, prompt risk stratification is needed to identify the patients with the highest probability of developing a more severe disease in order to provide the best clinical care and the best clinical setting for their management [10].

Biologically active adrenomedullin (Bio-ADM) is an emerging biomarker of endothelial function and vascular integrity already proven to be of utility in patients with sepsis in the ED. Plasma adrenomedullin is associated with short-term mortality and vasopressor requirement in patients admitted with sepsis [11].

Bio-ADM could serve as a useful and objective biomarker to predict severity, organ failure, and 30-day mortality in septic patients [10,12,13]. Histopathological studies conducted on the lung cells of dead patients with COVID-19 disease highlighted multifocal endotheliitis, endothelial damage, and angiogenesis, demonstrating the importance of the endothelium in the pathogenesis of COVID-19 [14].

Proenkephalin (penKid) is a promising biomarker that has been studied in critically ill patients which may play a role in the evolution of COVID-19 infection in a life-threatening disease and which has never been studied yet in the context of interstitial pneumonia [15]. The biomarker concentration appears to be inversely correlated with the glomerular filtrate rate (GFR) and therefore correlated with renal function [16]. Acute kidney injury (AKI) is a major complication of severe COVID-19 disease, caused by an uncontrolled systemic inflammatory response with a consequent risk of intrarenal inflammation [17]. PenKid seems to be an effective predictor of renal injury, severe multi-organ failure, and mortality in unselected sepsis patients admitted to the emergency ward [15,18].

The aim of this study was to investigate the prognostic role in short-term mortality of Bio-ADM and penKid in ED patients affected by interstitial pneumonia caused by COVID-19 in comparison with patients with interstitial pneumonia of any other cause.

## 2. Materials and Methods

This monocentric, prospective observational study was conducted in the ED of San Giovanni Addolorata Hospital on behalf of the postgraduate medical school of Emergency Medicine of the University Sapienza of Rome. The study was approved by the institutional Research Ethics Committee “Lazio 2” (Process N. 0157973/2020, 30.09.2020). After informed consent was signed, we enrolled 153 patients who were admitted to the emergency ward from April 2020 to April 2021.

Inclusion criteria:-Age ≥ 18 years;-First-presentation symptoms such as dyspnea, fever, and dry cough;-High-Resolution Computed Tomography (HRCT) scan findings of suspicious images for interstitial pneumonia;-Adhesion to informed consent.

Exclusion criteria:-Age < 18 years;-Refusal of consent;-Dyspnea from a non-infectious cause;-Symptom onset more than 3 days before the enrollment.

The HRCT was considered suspect for interstitial lung involvement if more than 25% of the lung parenchyma was involved or if ground glass, crazy paving, architectural distortion, honeycombing, or consolidations were found in more than two lung fields [19,20,21].

All of the patients enrolled underwent two SARS-CoV-2 oro-pharyngeal swabs within 24 h of each other to confirm or exclude SARS-CoV-2 infection. We used real-time polymerase chain reaction (RT-PCR) molecular tests and TaqMan-based detection (Taqman™ Real-Time PCR Assays-Thermo Fisher Scientific™, Frederick, MD 21704, USA).

### 2.1. Study Design

Enrolled patients were evaluated on ED admission (T0), after 24 h (T1), and then 30 days later (T30) with a telephone call follow-up.

At T0, a SARS-CoV-2 swab, lung HRCT, routine blood tests, and arterial blood gas analysis (ABG) were performed on each enrolled patient. Subjects who met the enrollment criteria were assigned to one of the two groups, COVID+ or COVID−, depending on whether they were, respectively, positive or negative on the swab. An additional blood sample was withdrawn through a vacutainer system with EthylenDiaminoTetracyc Acid (EDTA) and stored at −20 °C in a freezer for further analysis of Bio-ADM and penKid. Clinical history before admission to the ED, existing therapies, drug allergies, ongoing symptoms, radiological findings, and hemodynamic parameters were also recorded at this time. qSOFA and SOFA scores were recorded at T0 and recalculated at T1 to stratify patients’ mortality risk.

At T1, patients underwent a new clinical evaluation and qSOFA and SOFA were recalculated and recorded. Patients negative on the first oropharyngeal swap underwent a second test to confirm the negative results and, in case of positivity, were switched to the COVID+ group. We used RT-PCR molecular tests and TaqMan-based detection (Taqman™ Real-Time PCR Assays-Thermo Fisher Scientific™, Frederick, MD 21704, USA).

A second sample in an EDTA-containing vacutainer for Bio-ADM and penKid measurements, as well as another ABG, was obtained for all patients at this time.

At T30, patients were contacted by telephone for follow-up. The onset of any potential complications, new hospitalization, or death in the previous 30 days was recorded. Attention was paid to whether complications arose in the first 10 days after ED admission (T10) or in the following 30 days (T30).

### 2.2. Biomarkers Analysis

Both T0 and T1 venous samples were centrifuged within one hour in San Giovanni Addolorata Hospital with a swinging bucket centrifuge set at 1200 rpm for 10 min at 36 °C. We then obtained two 0.5 mL centrifuged samples and stored them at −20 °C. The obtained samples were shipped to Sphingotec GmbH laboratories in Germany for the analysis of Bio-ADM and penKid concentrations.

Plasma Bio-ADM was measured using an immunoassay (sphingotest^®^Bio-ADM^®^) developed by Sphingotec GmbH (Hennigsdorf, Germany). Bio-ADM immunoassay is a one-step sandwich chemiluminescence immunoassay based on acridinium NHS-ester labeling for the detection of human adrenomedullin (ADM) in unprocessed, neat plasma. It uses two mouse monoclonal antibodies, one directed against the mid-region and the other directed against the amidated C-terminal moiety of ADM. The assay uses 50 µL of plasma samples/calibrators and 220 µL of labeled detection antibody. Once thawed back to room temperature, Bio-ADM in EDTA plasma is stable for up to 24 h, and samples are unaffected by at least up to four freeze–thaw cycles. This assay is highly specific to Bio-ADM as it only reacts with the mature amidated C-terminus of ADM and not with other variants of the precursor of ADM (Pro-ADM). Pro-ADM was measured using a sandwich ELISA on a Luminex^®^platform (Alere Inc., San Diego, CA, USA) [22]. In healthy subjects, the median Bio-ADM concentration was 20.7 pg/mL (99th percentile: 43 pg/mL) [23].

PenKid was measured in duplicate using a chemiluminescence immunoassay (Sphingotec GmbH, Hennigsdorf, Germany), as described: 2 mouse monoclonal anti-PenK antibodies were developed by immunization with PENK peptide (amino acids 119 to 159 of proenkephalin A). One antibody (2 mg) was used to coat polystyrene tubes. The other antibody labeled with methylacridinium ester served as the detector antibody. Standards (PENK peptide; amino acids 119 to 159 of proenkephalin A) and samples (50 mL) were incubated in tubes with the detector antibody (150 mL). After equilibration, the tubes were washed, and bound chemiluminescence was detected with a luminometer [24].

### 2.3. Statistical Analysis

The patients were divided into two groups (COVID− and COVID+) according to the results of the swab. The statistical significance was determined with a threshold of 95% (α < 0.05). The whole statistical examination has been performed using the statistical free software R (version 3.4.3—R Foundation, Free Software Foundation’s GNU project). Continuous variables were tested for normality with the Shapiro–Wilk test and summarized as mean and standard deviation (SD).

The differences between the means of the different groups were tested using a single-way ANOVA and Tukey’s test for post hoc analysis. Categorical variables were summarized in crosstab, expressed as a% of the own group, and analyzed using a χ2 test. If the test gave a significant result, that was further analyzed with a z-test.

The box-plot graphs were realized for the main results in order to visualize the values of distribution. A survival analysis was performed in order to evaluate the predictive power of Bio-ADM and penKid. The cut-off for the variables utilized for the analysis was 100 pg/mL and 43 pmol/L, respectively, for penKid and Bio-ADM, as previously described by Lundberg OHM et al. [25] and Marino R et al. [16].

The differences in survival time between the groups were graphically compared by constructing Kaplan–Maier curves and verified with a log-rank test.

## 3. Results

### 3.1. Population Characteristics

We enrolled 153 patients (54.5% female): 101 (66%) affected by COVID-19 pneumonia (COVID+) and 52 (34%) affected by interstitial pneumonia not related to SARS-CoV-2 (COVID−). The mean age of the entire population was 70 ±17 years, with no differences between COVID+ and COVID− (respectively, 68 ± 17 vs. 72 ± 17, *p*-value = 0.2).

Patient characteristics are shown in Table 1.

Table 1 summarizes the demographic characteristics, presenting symptoms, patients’ home therapy, the degree of intensity of care, and outcomes such as death and the onset of complications. There were no differences in comorbidity between COVID+ and COVID− groups, as shown in Table 1. The most common symptom leading to admission to the ED was dyspnea, as it affected 90 patients (60%) without statistically significant differences between COVID+ and COVID-. The other more frequent symptoms were fever in 72 cases (47.1%), cough in 36 (23.5%), and other minor symptoms such as myalgias or diarrhea in 51 (33.3%), with more than one symptom in 73 patients (47.7%). Fever was more frequent in COVID+ than COVID− (respectively, 57.4% vs. 28.8%, *p*-value = 0.001).

A successful rate of response was obtained in 109 patients (71%) upon follow-up and the percentage of dropout was the same in the two groups. Among the 44 patients for whom we were unable to obtain follow-up, there were some who were homeless, some who were foreigners in transit in Rome, and some patients for whom the address we collected was no longer available at the time of the call. Fifteen patients (9.8%) of the entire studied cohort died within 30 days. Two deaths occurred in the first 24 h. Six patients died within the first 10 days. The number of deaths increased by another seven units at 30 days. Most deaths, 12 cases, occurred in the COVID+ group (11.9%), compared to COVID−, in which only three patients died (5.8%); however, the difference did not reach statistical significance. Both major and minor complications were more common in the COVID+ group compared to COVID-: respectively, 19.6% vs. 5.8% for major complications (*p*-value 0.053) and 19% vs. 5% for minor complications (*p*-value 0.12).

Table 1 also shows every clinical and laboratory variable of every patient, both COVID and non-COVID, upon admission to the ED. SOFA scores were strongly correlated with the cause of interstitial pneumonia. The mean value in the COVID+ group is 3 ± 3, while in COVID− it is 6 ± 3, with a *p*-value <0.0001. There were no differences in clinical parameters between the two groups, except for a slightly higher body temperature in the COVID+ group compared to COVID− (respectively, 36.9 ± 1.0 °C vs. 36.3 ± 0.6 °C, *p*-value = 0.0001).

No changes in blood gas exchanges were found between the two pulmonary interstitial infections with similar P/F values between COVID+ (359 ± 168) and COVID− (363 ± 134, *p*-value = 0.9). Hb and pH appeared higher in the COVID+ group (respectively, 13 ± 2 g/dL and 7.44 ± 0.06) than in COVID− (respectively, 12 ± 3 g/dL, *p*-value = 0.002, and 7.42 ± 0.05, *p*-value = 0.0089), with a slight but not significant reduction in Lac in the COVID+ group with respect to COVID− (1.8 ± 0.8 mmol/L vs. 2.1 ± 1.7 mmol/L, *p*-value = 0.08). These results are consistent with the worst oxygen delivery in the COVID− group at the time of presentation at the ED. In COVID+, there was a different cellular peripheral inflammatory response with a reduced white blood cell (WBC) count compared to COVID− (respectively, 6 ± 3 × 10^3^/uL vs. 12 ± 7 × 10^3^/uL, *p*-value < 0.0001) and an increased neutrophil concentration compared to COVID− (77 ± 13% vs. 67 ± 22%, *p*-value = 0.0195). Platelet count was also reduced in the COVID+ group compared to COVID− (193 ± 89 × 10^3^/uL vs. 259 ± 117 × 10^3^/uL, *p*-value = 0.0002). No other significant differences were found between the other laboratory tests.

### 3.2. Biomarkers and Etiology Correlation

Figure 1 shows the box-plot graph of Bio-ADM values (panel A) and penKid values (panel B) in COVID+ and COVID− groups at T0. We did not find any difference in Bio-ADM and penKid concentrations between COVID+ and COVID− at T0.

Bio-ADM values strongly correlated with mortality at T1 (*p*-value = 0.0004), T10 (*p*-value = 0.0003), and T30 (*p*-value = 0.003). The mean values of Bio-ADM at T0 were 184 ± 98 pg/mL for those who died within the first 24 h, 101 ± 71 pg/mL for those who died in the first 10 days, and 73 ± 60 pg/mL in patients who died at the 30th day of follow up. The respective values were in the group of survivors at T1: 37 ± 31 pg/mL; for the survivors at T10: 36 ± 29 pg/mL; and in the group of survivors at T30: 36 ± 30 pg/mL. These results are shown in Figure 2.

No relationship was found between Bio-ADM values and the development of complications both major and minor. In the first group of patients, the mean ± SD was 43.03 ± 45.63 pg/mL, and in the patients that developed complications, this value was 49.4 ± 32.89 pg/m (*p*-value 0.20).

PenKid values did not correlate with mortality at T1 and T10 but were predictive of mortality at T30; *p*-value = 0.02. The mean ± SD in the group of patients who died at 30 days was 99 ± 66 pmol/L compared to 55 ± 33 pmol/L of survivors, as shown in Figure 3.

The Kaplan–Meier curve (Figure 4a) demonstrates significantly higher survival rates in patients with Bio-ADM <43 pg/mL than patients with Bio-ADM ≥43 pg/mL, as well as in patients with penKid <100 pmol/L than patients with penKid ≥100 pmol/L (Figure 4b). The shape of the two survival curves separates very early when we use Bio-ADM as a predictor variable, and only in the final part when we use penKid, which could mean that Bio-ADM is a predictor of early mortality while penKid correlates only with global mortality and disease-induced complications.

We performed a Cox analysis, including values of Bio-ADM and PenKid over or below the cutoff at T0 in order to verify their Hazard Ratio (HR). We also decided to forcefully add age, creatinine value at T0, and COVID-19-positive diagnosis in the analysis, considering these variables as potentially predictive of mortality. The complete model results significantly differ from the null model (*p* 0.0001).

As shown in Table 2, the only two variables left in the model (Wald Test > 0) were age and BioADM.

With an HR of 2.95 (1.16–7.46 95%CI, *p* 0.02) for Bio-ADM > 43 pg/mL and an HR of 1.09 (1.05–1.14 95%CI, *p* 0.0001) for age.

As can be seen from Figure 5, penKid confirmed its predictive value for acute AKI. The mean ± SD in the group of patients who developed AKI was 166 ± 114 pmol/L vs. 61 ± 45 pmol/L in patients who did not develop it (*p*-value = 0.0013).

We observed a positive linear regression between PenKid levels at T0 and creatinine at T0 with an R of 0.55 and a *p*-value of 0.0001.

A correlation analysis between AKI and PenKid > 100 mmol/L was performed, and it showed a slight positive correlation with a *p*-value of 0.003 but an R of only 0.29 (probably due to the small number of AKIs that occurred; only four).

Neither Bio-ADM nor penKid appear to be influenced in any way by the etiology of lung infection and no differences were observed in biomarker predictivity in the COVID + and COVID− groups.

## 4. Discussion

The main finding of our study has been the predictive mortality value of Bio-ADM and penKid, measured in the ED in patients presenting with interstitial pneumonia both related and not related to COVID-19 infection. Furthermore, Bio-ADM seems to correlate mostly with short-term mortality (in hospital and within 10 days), while penKid correlates better with 30-day mortality.

In the absence of specific treatment strategies for COVID-19, there is an urgent call for clear guidance by early risk-stratifying biomarkers [26]. In our study, we enrolled two groups of patients: one with COVID-19 interstitial pneumonia and the other with SARS-CoV-2-negative interstitial pneumonia. Despite major differences between COVID and non-COVID interstitial pneumonia’s physiopathology and clinical evolution, we were not able to find any dissimilarity in Bio-ADM and penKid expression and predictive power. The two groups indeed did not show significant differences between COVID-19-positive and -negative patient biomarker values, suggesting that the two biomarkers are involved in a systemic inflammatory response independent of the mechanism that causes the inflammation.

As far as we know, this is the first study proving a strong correlation between admission Bio-ADM levels and mortality within 24 h (*p*-value 0.0004) in patients with interstitial pneumonia. According to the currently available data, there is no study proving a relationship between Bio-ADM values at ED admission and 24 h mortality in patients with interstitial or lobar pneumonia, sepsis, or septic shock. The earliest mortality studied so far is at 4 days [25]. The possibility of measuring a biomarker able to predict 24 h mortality could greatly improve the efficiency of medical staff working in the ED to identify and treat the most severe cases and take early clinical decisions impacting patient prognosis. Furthermore, in an emergency overcrowding setting such as the one caused by COVID-19, in which the ED was overburdened by a significant number of interstitial pneumonia cases, this prompt risk stratification assessment could prove to be of great aid. The prognostic role of Bio-ADM values higher than the cut-off we chose has also been confirmed for mortality within 10 and 30 days (*p*-value of 0.0003 and 0.0032, respectively). These results are comparable to those that have already been obtained in various studies already published in the literature which establish how Bio-ADM values higher than the cut-off correlate with mortality in septic patients [25,27]. However, those previous studies have been conducted on patients either with sepsis or septic shock and using 70 pg/dL as the Bio-ADM value cut-off [13,25,28,29]. Since our study has involved a very different population consisting of interstitial pneumonia patients instead of sepsis or septic shock, we chose as a Bio-ADM cut-off a value of 43 pg/dL, which is the 99th percentile of normal Bio-ADM values as derived from a trial which was comprising 200 healthy patients whose samples were analyzed for Bio-ADM values [23,25]. In the literature, Bio-ADM also correlates with the need for more intensive treatment, such as vasopressors, orotracheal intubation (OTI), and admission to the Intensive Care Unit (ICU). This correlation appears weaker in our data, likely because the populations under examination were so different [25,27]. We also observed that the SOFA score was strongly correlated with the cause of interstitial pneumonia (the COVID− group was higher, with a *p*-value < 0.0001). We may suppose that since the SOFA score assesses sepsis-related organ failure and sepsis is usually determined by bacteria, it was more likely for a patient suffering from non-COVID pneumonia to develop sepsis.

Bio-ADM could also prove useful as a therapeutic target: Adrecizumab (ADZ) is a humanized monoclonal antibody targeting the N-terminus of ADM [30,31,32], and it is being evaluated as a promising specific therapy for sepsis [32,33]. Although the direct involvement of Bio-ADM in the etiopathogenesis of interstitial pneumonia must yet be proven, its usefulness in predicting mortality within 24 h, 10 days, and 30 days could justify new studies aimed at testing ADZ as a therapeutic tool in this clinical condition. The failure in highlighting any differences between COVID-19 and non-COVID-19 pneumonia suggests that Bio-ADM predictivity should be independent of the etiology of sepsis and it should be an interesting issue for future research. It could be that the biomarker level increases the dosage regardless of the etiology of the systemic inflammatory response and the modulation of its activity could be useful in a wide spectrum of pathologies.

In the last few years, many studies concerning penKid have been carried out. Our objective was to evaluate the biomarker’s diagnostic and prognostic role as a predictive factor of AKI [34]. It is known that GFR may be overestimated using both GFR based on Modifications of Diet in Renal Disease (GFR-MDRD) and GFR based on Endogenous Creatinine Clearance (GFR-ECC), two methods widely used in clinical practice in order to assess kidney function in unstable septic patients; this leads to an erroneous stratification of a large number of patients, which, in turn, could result in a negative influence on the wrongly attributed dosage of drugs based on an overestimated GFR. The plasma concentration of penKid, instead, shows a closer relationship with actual GFR, which could prove its use as a more accurate and simpler marker of kidney function in unstable septic patients [35]. From our study results, it appears that penKid retains its capacity to also detect kidney damage early in interstitial pneumonia, even in patients who are neither septic nor unstable. Compared to other emergent biomarkers [35,36,37,38,39], penKid provides two advantages: firstly, it can be obtained from blood samples; secondly, the consistent recording of penKid values in the range of normality in septic patients without AKI seems to prove that it is not influenced by the inflammatory response, which could suggest its increased specificity against other biomarkers [40].

Many studies addressing the diagnostic and prognostic role of penKid in various types of patients both in the ED and ICU are available in the literature. It has been proven that elevated blood penKid values at admission to the ICU are associated with organ failure occurring within the first 2 days, they correlate with kidney dysfunction, and they are good predictors of 30-day mortality regardless of the disease leading to the hospitalization. Even though enrollment criteria were different, our study set in the ED confirmed this biomarker’s role in predicting renal impairment, and the early detection of this marker at admission to the ED could allow the early start of a therapy that optimizes renal function, perhaps reducing the onset of irreversible damage or disability. The fact that penKid predicts mainly 30-day mortality may be due to the fact that this biomarker is a good predictor of disease sequelae; this kind of predictive power has already been evidenced in a previous retrospective trial conducted on patients admitted to the ICU for many reasons ranging from sepsis, cardiac arrest, or trauma [41].

## 5. Conclusions

Our study shows that in patients presenting to the ED with either COVID+ or COVID− interstitial pneumonia, Bio-ADM seems to be a strong predictor of 24 h, 10-day, and 30-day mortality. PenKid strongly relates to AKI and mortality at 30 days.

Therefore, it seems that the usage of these two biomarkers in the setting of the ED for prompt risk stratification could lead to a better and faster decision-making process and could improve the prognosis of this kind of patient.

## Figures and Tables

**Figure 1 medicina-58-01852-f001:**
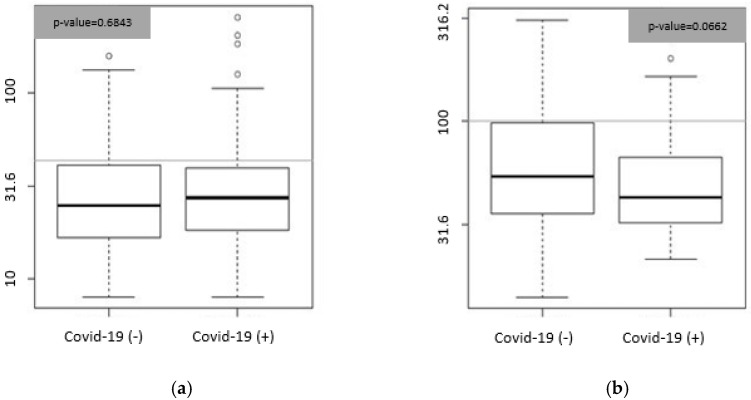
(**a**) Box plot. Concentrations expressed in pg/mL of Bio-ADM in the COVID– (*n* = 52) and COVID+ group (*n* = 101)) at T0. Gray lines represent the 99th percentile of normal values (43 pg/mL); black lines represent median values (COVID− = 24.8; COVID+ = 27.3). (**b**) Box plot. Concentrations expressed in pmol/L of penKid in the COVID− (*n* = 52) and COVID+ group (*n* = 101). Gray lines represent the 99th percentile of normal values (100 pmol/L); black lines represent median values (COVID− = 54.14; COVID+ = 43).

**Figure 2 medicina-58-01852-f002:**
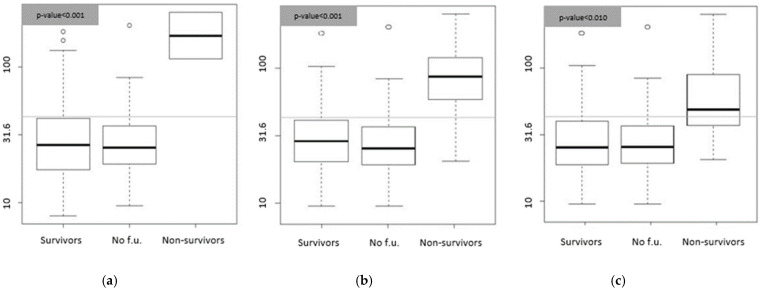
Bio-ADM concentrations expressed in pg/mL. (**a**) Box plot. Survivors: distribution of the Bio-ADM values of the patients who survived at 24 h (*n* = 107, median = 26.7); no f.u.: distribution of Bio-ADM values in patients for whom follow-up is not available (*n* = 44, median = 25.5); non-survivors: distribution of the Bio-ADM values of the patients who were dead within 24 h (*n* = 2, median = 183.8). (**b**) Box plot. Survivors: distribution of the Bio-ADM values of the patients who survived after 10 days (*n* = 101, median 28.9); no f.u.: distribution of Bio-ADM values in patients for whom follow-up is not available (*n* = 44, media = 25.5); non-survivors: distribution of the penKid values of the patients who were dead after 10 days (*n* = 8, median 88.5). (**c**) Box plot. Survivors: distribution of the Bio-ADM values of the patients who survived after 30 days (*n* = 94, median 25.3); no f.u.: distribution of Bio-ADM values in patients for whom follow-up is not available (*n* = 44 median 25.3); non-survivors: distribution of the penKid values of the patients who were dead after 30 days (*n* = 15, median = 48.7). Gray lines represent the 99th percentile of normal values (43 pg/mL).

**Figure 3 medicina-58-01852-f003:**
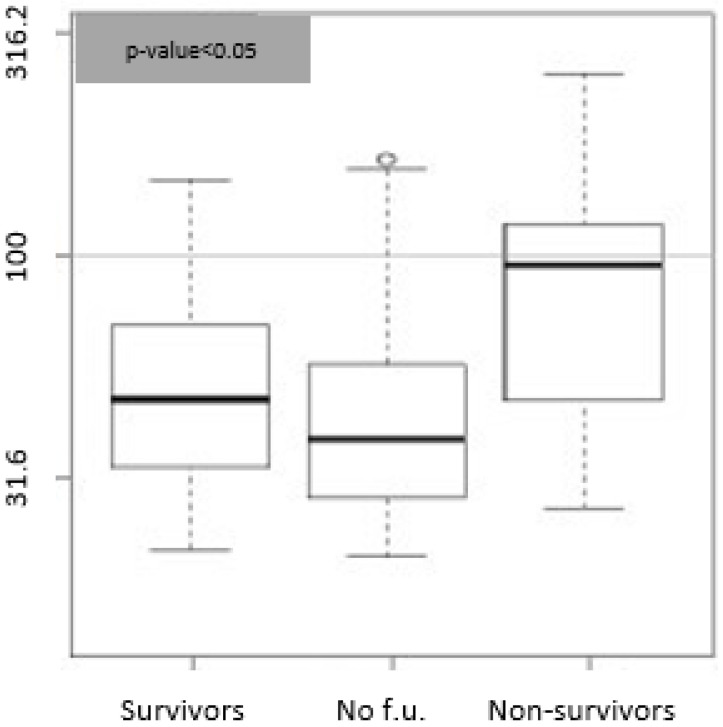
PenKid concentrations expressed in pmol/L. Box plots. Survivor: distribution of the penKid values of the patients who survived after 30 days (*n* = 94, median = 47.5); no f.u.: distribution of penKid values in patients for whom follow-up is not available (*n* = 44, median = 42.3); non-survivors: distribution of the penKid values of the patients who were dead after 30 days (*n* = 15, median = 95.1). Gray lines represent the 99th percentile of normal values (100 pmol/L).

**Figure 4 medicina-58-01852-f004:**
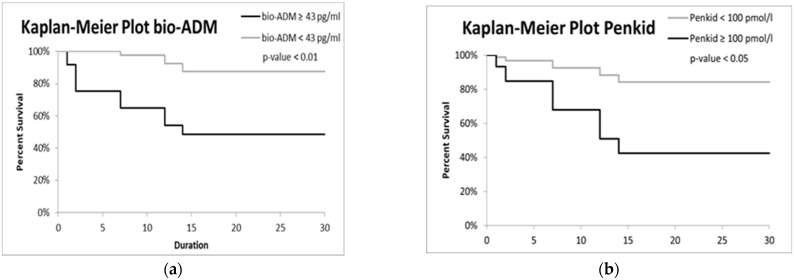
(**a**) Kaplan–Meier Bio-ADM curve. Grey line shows the survival rate for the patients with Bio-ADM values < 43 pg/mL in the first 30 days. Black line shows the survival rate for the patients with Bio-ADM values ≥ 43 pg/mL (*p*-value 0.003). (**b**) Kaplan–Meier penKid curve. Grey line shows the survival rate for the patients with penKid values < 100 pmol/L in the first 30 days. Black line shows the survival rate for the patients with penKid values ≥ 100 pmol/L (*p*-value 0.02).

**Figure 5 medicina-58-01852-f005:**
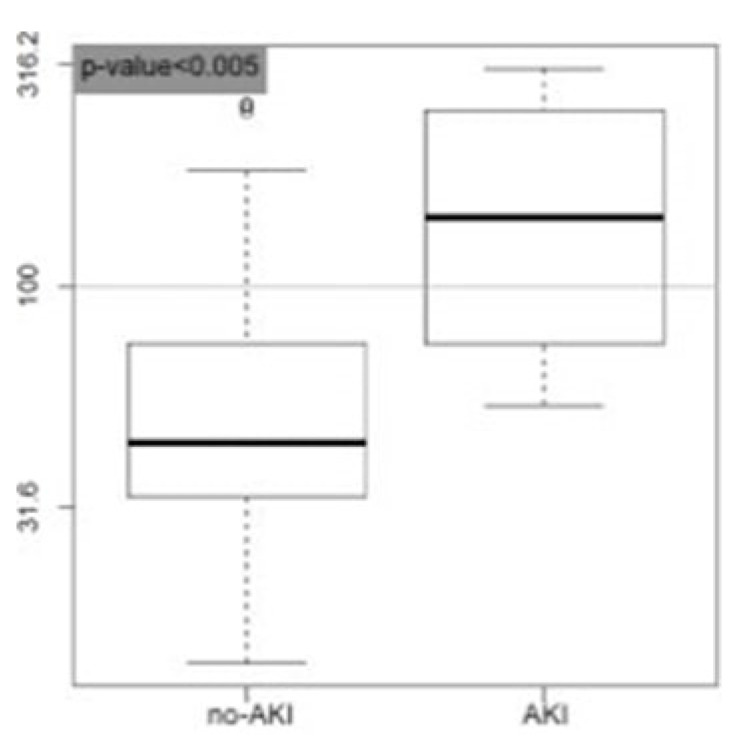
PenKid concentrations expressed in pmol/L. Box plots. No-AKI: distribution of the penKid values of the patients who did not develop AKI (*n* = 149, median = 44.3); AKI: distribution of the penKid values of the patients who developed AKI (*n* = 4, median = 151.1).

**Table 1 medicina-58-01852-t001:** Patient characteristics. Abbreviations: IHD, Ischemic Heart Disease; CKD, Chronic Kidney Failure; COPD, Chronic Obstructive Pulmonary Disease; RR, respiratory rate; MBP, mean blood pressure; HR, heart rate; BT, body temperature; WBCs, white blood cell; CPR, C-Protein Reactive; PCT, Procalcitonin; HSTn, High-Sensitivity Cardiac Troponin; BNP, brain natriuretic peptide.

	All (153)	COVID+ (101)	COVID− (52)	*p*-Value
DEMOGRAPHICS				
Age (mean ± SD)	70 ± 17	68 ± 17	72 ± 17	0.2
Female n. (%)	69 (45.1%)	46 (45.5%)	23 (44.2%)	0.88
COEXISTHING CONDITIONS n. (%)				
Hypertension	69 (45.1%)	46 (44.5%)	23 (44.2%)	0.88
Diabetes	24 (15.7%)	14 (13.9)	10 (19.2%)	0.38
IHD	11 (7.2%)	7 (6.9%)	4 (7.7%)	0.86
CKD	6 (3.9%)	3 (3.0%)	3 (5.7%)	0.39
COPD	19 (12.4%)	9 (8.9%)	10 (19.2%)	0.07
SYMPTOMS OF PRESENTATION IN ED n. (%)				
Dyspnea	90 (60.1%)	61 (60.4%)	31 (59.6%)	0.92
Cough	36 (23.5%)	26 (25.4%)	10 (19.2%)	0.36
Fever	72 (47.1%)	58 (57.4%)	14 (26.9%)	0.001
Other	51 (33.3%)	38 (37.6%)	13 (25.5%)	0.11
More than one	73 (47.7%)	58 (57.4%)	15 (28.8%)	0.0008
DEATH n. (%)	15 (9.8%)	12 (11.9%)	3 (5.8%)	0.22
COMPLICATIONS n. (%)				
Major complications	23 (14.6%)	20 (19,6%)	3 (5.8%)	0.053
AKI	4 (2.6%)	3 (2.97%)	1 (1.92%)	0.70
Light complications	19 (12.4%)	16 (15.8%)	3 (5.8%)	0.15
DESTINATION n. (%)				
Low intensive care unit	77 (50.3%)	63 (62.4%)	14 (26.9%)	0.00003
Sub-intensive therapy	73 (47.7%)	36 (35.6%)	37 (71.2%)	0.00003
Intensive care unit	3 (2.0%)	2 (2%)	1 (0.9%)	0.98
HOME THERAPY n. (%)				
Oral anticoagulants	17 (11.1%)	12 (11.9%)	5 (9.4%)	0.70
Antiplatelets	30 (19.6%)	17 (16.8%)	13 (25%)	0.22
Antihypertensive	59 (38.6%)	42 (41.6%)	17 (32.7%)	0.28
Cortisone	9 (5.9%)	5 (5%)	4 (7.7%)	0.49
Oral hypoglycemic	17 (11.1%)	12 (11.9%)	5 (9.6%)	0.70
Insulin	8 (5.2%)	6 (5.9%)	2 (3.8%)	0.58
Anti-arrhythmic	21 (13.7%)	17 (16.8%)	4 (7.7%)	0.12
Chemotherapy	2 (1.3%)	1 (1.0%)	1 (1.9%)	0.63
Diuretics	29 (19%)	19 (18.8%)	10 (19.2%)	<0.05
SOFA	4 ± 3	3 ± 3	6 ± 3	<0.001
RR (b.p.r)	20 ± 5	20 ± 5	21 ± 6	0.31
SpO2 (%)	94 ± 6	94 ± 5	94 ± 7	0.18
MBP (mmHg)	95 ± 15	95 ± 13	95 ± 19	0.9
HR (b.p.m)	87 ± 18	87 ± 20	86 ± 14	0.74
BT (°C)	36.7 ± 0.9	36.9 ± 1.0	36.3 ± 0.6	<0.001
P/F (mmHg/%)	360 ± 156	359 ± 168	363 ± 134	0.9
Hb (g/dL)	13 ± 2	13 ± 2	12 ± 3	<0.01
pH	7.44 ± 0.06	7.44 ± 0.06	7.42 ± 0.05	<0.01
Lac (mmol/L)	1.9 ± 1.2	1.8 ± 0.8	2.1 ± 1.8	0.08
WBCs (×10^3^/uL)	8 ± 5	6 ± 3	12 ± 7	<0.001
Neutrophils (%)	76 ± 14	78 ± 13	67 ± 22	<0.05
Plt (×10^3^/uL)	215 ± 104	193 ± 90	259 ± 117	<0.001
CRP (ng/mL)	7 ± 7	7 ± 7	7 ± 8	0.9
PCT (ug/mL)	1 ± 5	1 ± 6	1 ± 5	0.9
Creat (mg/dL)	1.1 ± 0.6	1.0 ± 0.5	1.2 ± 0.7	0.05
Bilirubin Tot. (mg/dL)	0.9 ± 1.6	0.8 ± 0.6	1.2 ± 2.6	0.14
INR	1.3 ± 0.8	1.2 ± 0.5	1.4 ± 1.2	0.14
LDH (U/L)	356 ± 219	357 ± 177	354 ± 288	0.95
K (mmol/L)	3.7 ± 0.5	3.9 ± 0.5	4.1 ± 0.6	0.05
Na (mmol/L)	137 ± 4	137 ± 4	138 ± 6	0.22
HSTn (ng/L)	316 ± 1866	257 ± 661	426 ± 2213	0.21
BNP (pg/mL)	313 ± 657	233 ± 612	481 ± 724	0.05

**Table 2 medicina-58-01852-t002:** Cox analysis for Bio-ADM, penKid, creatinine, COVID-19 positivity, and age.

Covariate	Beta	Sdt. Err.	Wald	*p*-Value	HR	*LCL*	*UCL*
Bio-ADM > 43 pg/mL	1.08	0.47	5.21	0.02	2.95	1.16	7.46
PenKid > 100 mmol/L	0.02	0.47	0.00	0.96	1.02	0.41	2.55
Creat_T_0_	0.003	0.17	0.00	0.98	1.00	0.72	1.39
COVID+	0.37	0.40	0,86	0.35	1.45	0.66	3.18
Age, years	0.09	0.02	15.96	0.0001	1.09	1.05	1.14

## Data Availability

The data presented in this study are available on request from the corresponding author. The data are not publicly available due to patient confidentiality.

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
