# Peer review of "Utility of Measuring Circulating Bio-Adrenomedullin and Proenkephalin for 30-Day Mortality Risk Prediction in Patients with COVID-19 and Non-COVID-19 Interstitial Pneumonia in the Emergency Department"

_medicina, 2022, doi:10.3390/medicina58121852_

Round 1
Reviewer 1 Report
1. Reference 11: Please cite the reference correctly (authors, journal name, issue number, page number, etc).
2. Introduction: “ … more than 554 million cases of Covid-19 have been confirmed as of 12th of July 2022 with more than 6 million deaths[2].”. Please update the data to more recent one.
3. Emergency Department vs. ED: the full term and abbreviation are used randomly throughout the text. It is not confined to ED. Other terms are also mixed (for example, GFR). Please check the correct use of abbreviations.
4. Method (page 3): Was the sample storage temperature -80 ℃ or -20 ℃? The two temperatures are described in the Methods. Please check!
5. Reference 22: It is for bio-ADM, but in the main text, it is cited for penKid (page 4). Please check!
6. Figure 1, 2, 3, and 5: Instead of 0/1, no/yes, please provide the exact information (that is included in the figure legends). In the x axis, Please, use the terms Covid-19 (+), CoVid-19 (-), survivor, non-survivors, AKI, or no-AKI, etc. Moreover, please provide each P value in each figure.
7. Figure 4: Please provide numerical data on statistical significance. It is not suggested in the figure and the text.
8. Please merge Table 1 and Table 2 into one Table.
9. Figure 5: There is no data on how many patients developed AKI in Table 1.
10. Are there no authors for reference 27?
11. Abstract: “… to evaluate the diagnostic and prognostic role of two biomarkers”. How about deleting “diagnostic” from this sentence? This study is focused on the prognostic value of the biomarkers.
12. All the continuous data are presented as mean & SD. Were they all distributed normally?
Author Response
Reviewer #1
1. Reference 11: Please cite the reference correctly (authors, journal name, issue number, page number, etc).
Thank you for the observation; we corrected the reference.
- Introduction: “ … more than 554 million cases of Covid-19 have been confirmed as of 12th of July 2022 with more than 6 million deaths[2].”. Please update the data to more recent one.
Thank you for the remark. We updated the data and added the reference. - Emergency Department vs. ED: the full term and abbreviation are used randomly throughout the text. It is not confined to ED. Other terms are also mixed (for example, GFR). Please check the correct use of abbreviations.
Thank you. As far as we are aware of we corrected every single instance of this in the text.
4. Method (page 3): Was the sample storage temperature -80 ℃ or -20 ℃? The two temperatures are described in the Methods. Please check!
The storage temperature was indeed -20 °C; we corrected it. Thanks.
- Reference 22: It is for bio-ADM, but in the main text, it is cited for penKid (page 4). Please check!
Corrected in the text. Thanks.
- Figure 1, 2, 3, and 5: Instead of 0/1, no/yes, please provide the exact information (that is included in the figure legends). In the x axis, Please, use the terms Covid-19 (+), CoVid-19 (-)survivor, non-survivors, AKI, or no-AKI, etc. Moreover, please provide each P value in each figure.
Here are shown the corrected figures as they now appear in the paper.
- Figure 4: Please provide numerical data on statistical significance. It is not suggested in the figure and the text.
Thank you for the remark, we corrected the text.
- Please merge Table 1 and Table 2 into one Table.
Corrected in the text, thanks
- Figure 5: There is no data on how many patients developed AKI in Table 1.
Thank you for the observation, we corrected table 1.
- Are there no authors for reference 27?
We corrected the reference.
- Abstract: “… to evaluate the diagnostic and prognostic role of two biomarkers”. How about deleting “diagnostic” from this sentence? This study is focused on the prognostic value of the biomarkers.
Thank you for your advice; we changed the sentence as you suggested.
- All the continuous data are presented as mean & SD. Were they all distributed normally?
As specified in statistical analysis, the variables were tested for normality with a Shapiro-wik test (page 4, lines 27-28)
Reviewer 2 Report
This study investigated the clinical utility of Bio-ADM and penKid for predicting the prognosis of COVID-19 interstitial pneumonia and idiopathic interstitial pneumonia. This study can provide a useful tool to clinicians in ED, especially in terms of mortality prediction. However, I think the manuscript should be majorly revised to draw a definite conclusion that the authors suggested.
Major comments:
Materials and Methods
1. This study compared levels of Bio-ADM and penKid between COVID-19 interstitial pneumonia and non-COVID-19 interstitial pneumonia. This study enrolled patients with interstitial pneumonia by HRCT and excluded patients with dyspnea from a non-infectious cause. Are these criteria appropriate for enrolling all interstitial pneumonia patients? Many patients with idiopathic interstitial pneumonia do not have any infection evidence. Thus, did this study compare interstitial pneumonia caused by SARS-CoV-2 infection with interstitial pneumonia caused by other bacteria or viruses? If this is right, you should present the results for the infection causes identified in the non-COVID-19 group interstitial pneumonia in table 1.
2. In this study, the most critical points were to validate the clinical usefulness of Bio-ADM and penKid in COVID-19 interstitial pneumonia and non-COVID-19 interstitial pneumonia and to present a cut-off that can differentiate the two groups. The authors suggested cut-offs using previous references. (1) Reference 23 showed that the median Bio-ADM at baseline was 33.8 (22.6 – 53.9) pg/ml in heart failure suspected patients. Reference 23 should be changed to another reference (Marino et al. doi: 10.1186/cc13731). (2) Reference 24 did not state that 100 pmol/L is the proper cut-off. Reference 24 suggested cutoff values for the prediction of a high risk of acute myocardial infarction. Why did you choose 100 pmol/L as a cutoff for this study? (3) If there is no difference in Bio-ADM and penKid levels between COVID-19 interstitial pneumonia and non-COVID-19 interstitial pneumonia, it would be good to insert the results suggesting cutoffs for predicting mortality in both groups. (4) You should correct the following sentence in the Materials and Methods part: the cut off of the variables utilized for the analysis were 100 pg/ml and 43 pmol/l respectively for Bio-ADM and penKid.
3. The authors represented “n.s.” for statistically insignificant results in the tables, which caused confusion as specific p values were described in the Results part. It is better to indicate the p values in the tables as specific numbers for readability.
4. What is the evidence that SOFA score was strongly correlated with the cause of interstitial pneumonia? It is known that penKid levels increase as the SOFA score increase. The SOFA scoring system, which assesses sepsis-related organ failure, was significantly higher in the non-COVID-19 interstitial pneumonia group than in COVID-19 interstitial pneumonia. Please describe how you interpret this result in the discussion part.
5. The result of temperature in the COVID-19 interstitial pneumonia group is 36.9±1.0°C. What is the definition of fever in your study? In Table 1, 57.4% of the COVID-19 interstitial pneumonia patients and 26.9% of non-COVID-19 interstitial pneumonia patients had a fever, so it is necessary to check whether the mean BP was 36.9 ± 1.0. In addition, did the results of 36.9±1.0 and 36.3±0.6 in both groups really show a p-value <0.001 with statistical significance?
6. Figure 2 and Table 3 showed the same results. You have better remove the figure or table.
7. What is the evidence for no relationship between Bio-ADM and the development of major and minor complications?
8. Figure 3 and Table 3 showed the same results. You have better remove the figure or table.
9. In figure 4, the author showed that the mortality rate was high in the groups with high Bio-ADM and penKid levels. Although the log-rank test was performed, p values did not present in the figure. In addition, to analyze the factors such as Bio-ADM and penKid that affect mortality, the cox-regression multivariate analysis should be performed.
10. Author described that neither Bio-ADM nor penKid appears to be influenced in any way by the etiology of lung infection and no differences were observed in biomarker predictivity in the COVID+ and COVID- groups. In order to clearly describe this, it is necessary to present the results between Bio-ADM and penKid levels at T1, T10, and T30 as well as the difference at T0 shown in figure 1.
11. Figure 5 indicated that the AKI group showed high penKid levels as same as in previous studies. To approve the association between AKI and penKid level in interstitial pneumonia patients of ED, univariate and multivariate regression analysis should be performed.
Minor comments
1. Use the official name suggested by WHO uniformly. e.g. COVID-19, SARS-CoV-2
2. Please specify how many patients you enrolled in the Materials and Methods part.
3. For SASR-CoV-2 confirmatory test, please specify in detail which company’s reagent was used.
4. According to the description of the Materials and Methods part, additional blood collection for biomarker analyses was performed at T0 and T1. Those samples were sent to laboratories for Bio-ADM and penKid measurement. The results of T1, T10, and T30 were shown in the results part. Please correct or explain the differences.
5. What do you mean by “In Table 2 all covariates related with the two different etiologies of interstitial pneumonia are shown at T0). It would be better to correct the sentence to make it clear.
6. The result of P/F in the COVID-19 interstitial pneumonia group is 358±168. Correct it in the Result part.
7. In figure 2 legend, the number of survivals at T3 is not 107. In the figure 3 legend, the number of survivals at T3 is not 44.
8. Please check your references according to the journal format. In addition, several references missed some information (ref 5,11,36).
9. You should carefully check the English grammar and typo. e.g. dyspnoea, C°, La-, 103, etc. Plus, you should use ‘comma’ and ‘period’. Differently.
Author Response
Reviewer #2
Major comments:
Materials and Methods
- This study compared levels of Bio-ADM and penKid between COVID-19 interstitial pneumonia and non-COVID-19 interstitial pneumonia. This study enrolled patients with interstitial pneumonia by HRCT and excluded patients with dyspnea from a non-infectious cause. Are these criteria appropriate for enrolling all interstitial pneumonia patients? Many patients with idiopathic interstitial pneumonia do not have any infection evidence. Thus, did this study compare interstitial pneumonia caused by SARS-CoV-2 infection with interstitial pneumonia caused by other bacteria or viruses? If this is right, you should present the results for the infection causes identified in the non-COVID-19 group interstitial pneumonia in table 1.
Thank you for your remark. We enrolled only infective interstitial pneumonia patients; in the inclusion criteria (page 2) we considered acute dyspnea (< 3 days), not associated with non-infective causes, like pulmonary edema, chronic pulmonary disease, and vascular pulmonary embolism. We didn’t look in the etiological causes because the study was conducted in the ED, where there is no possibility to test other viral or bacterial agents in a short time. This is usually delegated to the ward of destination.
- In this study, the most critical points were to validate the clinical usefulness of Bio-ADM and penKid in COVID-19 interstitial pneumonia and non-COVID-19 interstitial pneumonia and to present a cut-off that can differentiate the two groups. The authors suggested cut-offs using previous references. (1) Reference 23 showed that the median Bio-ADM at baseline was 33.8 (22.6 – 53.9) pg/ml in heart failure suspected patients. Reference 23 should be changed to another reference (Marino et al. doi: 10.1186/cc13731). (2) Reference 24 did not state that 100 pmol/L is the proper cut-off. Reference 24 suggested cutoff values for the prediction of a high risk of acute myocardial infarction. Why did you choose 100 pmol/L as a cutoff for this study? (3) If there is no difference in Bio-ADM and penKid levels between COVID-19 interstitial pneumonia and non-COVID-19 interstitial pneumonia, it would be good to insert the results suggesting cutoffs for predicting mortality in both groups. (4) You should correct the following sentence in the Materials and Methods part: the cut off of the variables utilized for the analysis were 100 pg/ml and 43 pmol/l respectively for Bio-ADM and penKid.
Thank you for your attention;
1) We corrected the references.
2) We corrected the reference, cut-off is 43 pmol/L
3) We choose the two cut-off according to general literature. However, As you suggested we check the accuracy of that cut-off on our population, performing a ROC for bio-ADM and PenKid.
- the analysis gave us a cut-off of 43,4 ng/ml for bio-ADM (Se 0.62, Sp 0.79, AUC 0.67±07), almost the same of what already described in literature.
- The best cut-off value for PenKid in this population was 67,4 pmol/l (Se 0.67, Sp 0.65, AUC 0.63±08; accuracy 0.65). Nevertheless, the cut-off value of 100 pmol/l did not change significantly the predictivity of mortality in the KM analysis and it still preserved an high accuracy and acceptable Se and Sp values (Se 0.52, Sp 0.78, Accuracy 0.72). Therefore, we decided to use as a cut-off what has already been suggested in the previous studies.
4) We corrected the sentence.
- The authors represented “n.s.” for statistically insignificant results in the tables, which caused confusion as specific p values were described in the Results part. It is better to indicate the p values in the tables as specific numbers for readability.
Thank you. We corrected the tables.
- What is the evidence that SOFA score was strongly correlated with the cause of interstitial pneumonia? It is known that penKid levels increase as the SOFA score increase. The SOFA scoring system, which assesses sepsis-related organ failure, was significantly higher in the non-COVID-19 interstitial pneumonia group than in COVID-19 interstitial pneumonia. Please describe how you interpret this result in the discussion part.
Thank you. While we have not clear data on why the SOFA score is higher in non-COVID-19 interstitial pneumonia patients than in those affected by COVID-19 interstitial pneumonia, we hypothesize that since sepsis is usually caused by bacterial infection it is more likely for a patient who is already affected by a bacteria to undergo sepsis than one affected by a virus.
- The result of temperature in the COVID-19 interstitial pneumonia group is 36.9±1.0°C. What is the definition of fever in your study? In Table 1, 57.4% of the COVID-19 interstitial pneumonia patients and 26.9% of non-COVID-19 interstitial pneumonia patients had a fever, so it is necessary to check whether the mean BP was 36.9 ± 1.0. In addition, did the results of 36.9±1.0 and 36.3±0.6 in both groups really show a p-value <0.001 with statistical significance?
Thank you for noticing this discrepancy. We chose 37.5 °C as cut-off BT for fever (as suggested by WHO in COVID-19 patients). In table 1, “Patients characteristics” in particular in “SYMPTOMS OF PRESENTATION IN ED”, we described the symptoms for wich they were admitted in the ED; even if the fever was the main symptom, their actual body temperature could be lower than the cut-off we have chosen for different reasons (i.e., antipyretic drugs).
- Figure 2 and Table 3 showed the same results. You have better remove the figure or table. Thank you; we removed Table 3.
7. What is the evidence for no relationship between Bio-ADM and the development of major and minor complications?
Thank you for your remark. We didn’t find any significant difference in Bio-ADM values between the population that developed complications and the population that didn’t. We added statistic results in the text (page 7, lines 3-6)
- Figure 3 and Table 3 showed the same results. You have better remove the figure or table.
Thank you for the advice. We removed Table 3.
In figure 4, the author showed that the mortality rate was high in the groups with high Bio-ADM and penKid levels. Although the log-rank test was performed, p values did not present in the figure. In addition, to analyze the factors such as Bio-ADM and penKid that affect mortality, the cox-regression multivariate analysis should be performed.
Thank you, we corrected figure 4.
Thank you for your suggestion. We performed the cox analysis including in the analysis values of bioADM and PenKid over or below the cutoff a t0 in order to verify their HR. We decided to forcefully add also Age, Creatinine value at T0 and Positivity to COVID-19 diagnosis in the analysis, considering that variables as potentially predictive of mortality. The complete model results significant different form the null one (p 0.0001).
As shown in the following table, the only two variables left in the model (Wald Test > 0) were: Age and BioADM +
|
Covariate |
Beta |
Sdt. Err. |
Wald |
p-value |
HR |
LCL |
UCL |
|
Bio-ADM > 43 pg/ml |
1,08 |
0,47 |
5,21 |
0,02 |
2,95 |
1,16 |
7,46 |
|
PenKid > 100 mmol/l |
0,02 |
0,47 |
0,00 |
0,96 |
1,02 |
0,41 |
2,55 |
|
Creat_T0 |
0,003 |
0,17 |
0,00 |
0,98 |
1,00 |
0,72 |
1,39 |
|
COVID + |
0,37 |
0,40 |
0,86 |
0,35 |
1,45 |
0,66 |
3,18 |
|
Age, yrs |
0,09 |
0,02 |
15,96 |
0,0001 |
1,09 |
1,05 |
1,14 |
With a HR of 2.95 (1.16-7.46 95%CI, p 0.02) for Bio-ADM > 43 pg/ml and an HR 1.09 (1.05-1.14 95%CI, p 0.0001) for Age.
Now we add this observation in the text (page 9, lines 1-14)
- Author described that neither Bio-ADM nor penKid appears to be influenced in any way by the etiology of lung infection and no differences were observed in biomarker predictivity in the COVID+ and COVID- groups. In order to clearly describe this, it is necessary to present the results between Bio-ADM and penKid levels at T1, T10, and T30 as well as the difference at T0 shown in figure 1.
Thank you. For our study we chose to focus on the admission values of bio-ADM and penKid.
On T10 and T30 we evaluated only the eventual death of the patient or the development of any complication; maybe we were misleading in naming those two times like T0 and T1 and thus we could have avoided that.
- Figure 5 indicated that the AKI group showed high penKid levels as same as in previous studies. To approve the association between AKI and penKid level in interstitial pneumonia patients of ED, univariate and multivariate regression analysis should be performed.
Thank you for your observation. Aki group shown a high value of PenKid a To, as already described in the paper. in addition to this observation, we calculated the linear regression between Creatinine a T0 and PenKid a T0 in these patients. We observed a positive linear regression between PenKid levels a T0 and Creatinine a T0with a R 0.55 , p-value 0.0001 , as shown in the following figure.
In our study AKI was defined as the worsening of creatinine level or diuresis in the 24hr following the access to the emergency room. This worsening of renal function was observed in only 4 patients. A correlation between AKI and PenKid > 100 mmol/l was performed and it shown a slight positive correlation with a p-value of 0.003 but a R of only 0.29 (probably due to the small number of AKI occurred).
This is now more clearly described in the text (Page 9, Lines 24-28)
Minor comments
- Use the official name suggested by WHO uniformly. e.g. COVID-19, SARS-CoV-2
Thank you, we corrected it in the text.
- Please specify how many patients you enrolled in the Materials and Methods part.
Thank you, we added that.
- For SASR-CoV-2 confirmatory test, please specify in detail which company’s reagent was used.
We used RT-PCR molecular tests TaqMan-based detection (Taqman™ Real-Time PCR Assays-Thermo Fisher Scientific™, Frederick, Maryland, 21704, USA); we corrected the text (page 3, lines 13-14).
- According to the description of the Materials and Methods part, additional blood collection for biomarker analyses was performed at T0 and T1. Those samples were sent to laboratories for Bio-ADM and penKid measurement. The results of T1, T10, and T30 were shown in the results part. Please correct or explain the differences.
Thank you. We removed Table 3 in which those results were shown.
- What do you mean by “In Table 2 all covariates related with the two different etiologies of interstitial pneumonia are shown at T0). It would be better to correct the sentence to make it clear.
Thank you for your suggestion. We corrected the sentence. - The result of P/F in the COVID-19 interstitial pneumonia group is 358±168. Correct it in the Result part.
We corrected the text.
- In figure 2 legend, the number of survivals at T3 is not 107. In the figure 3 legend, the number of survivals at T3 is not 44.
We corrected the text, thanks.
- Please check your references according to the journal format. In addition, several references missed some information (ref 5,11,36).
Thank you. We corrected the references.
- You should carefully check the English grammar and typo. e.g. dyspnoea, C°, La-, 103, etc. Plus, you should use ‘comma’ and ‘period’. Differently.
Thank you for the advice. We corrected every mistake we found.
Reviewer 3 Report
Dear Authors,
Congratulations on your interesting and valuable research. I am looking forward to the results being incorporated in ER care for COVID patients. I also think these results may be of use in other pneumonias where endotheliitis is a potent pathogeneic factor.
I have a few questions:
The HRCT was considered suspect for interstitial lung involvement if more than 25% of the lung parenchyma was involved or if ground glass, crazy paving, architectural distortion, honeycombing or consolidations were found in more than two lung fields" - arch. distortion, honeycombing are considered end-stage fibrosis features, which can occur in any ILD. Have you included patients with only features of end-stage fibrosis or did you also require presence of active interstitial inflammation, e.g. GGO?
Exclusion criteria mentions dyspnea resulting from non-infectious causes; however, the lesions mentioned above may lead to dyspnea, moreover, the table also mentions comorbidities like COPD. Did you exclude patients with AEs of COPD or exacerbation of chronic circulatory diseases?
Author Response
REV 3
The HRCT was considered suspect for interstitial lung involvement if more than 25% of the lung parenchyma was involved or if ground glass, crazy paving, architectural distortion, honeycombing or consolidations were found in more than two lung fields" - arch. distortion, honeycombing are considered end-stage fibrosis features, which can occur in any ILD. Have you included patients with only features of end-stage fibrosis or did you also require presence of active interstitial inflammation, e.g. GGO?
We excluded every patient affected by non-infective interstitial pneumonia at the admission in ED.
Exclusion criteria mentions dyspnea resulting from non-infectious causes; however, the lesions mentioned above may lead to dyspnea, moreover, the table also mentions comorbidities like COPD. Did you exclude patients with AEs of COPD or exacerbation of chronic circulatory diseases?
Yes; we excluded every patient that didn’t show typical features of an ongoing infective process (time of onset, elevated inflammation indices, concomitant fever).
Round 2
Reviewer 2 Report
The authors responded and modified the manuscript to the reviewer’s comments accordingly. I have a few minor suggestions for the authors.
1. page 6, line 51. The data described in the results section was different from the data in figure 2. The mean values in the figure showed less than 31.6 pg/mL. So, you should check the data and the figure.
2. Figure 4. According to the figure legend, the grey line indicated penKid group with less than 100 pmol/L, and the black line indicated penKid group with more than 100 pmol/L. But, in figure 4, the colors for each group were reversed. You should modify the figure.
3. Throughout the manuscript, the authors have used both ‘comma’ and ‘period’ for the decimal point. Please change them all to periods.
Author Response
- page 6, line 51. The data described in the results section was different from the data in figure 2. The mean values in the figure showed less than 31.6 pg/mL. So, you should check the data and the figure.
Thank you. In the box-plot the black lines represented the median value. We corrected the figure’s legenda.
- Figure 4. According to the figure legend, the grey line indicated penKid group with less than 100 pmol/L, and the black line indicated penKid group with more than 100 pmol/L. But, in figure 4, the colors for each group were reversed. You should modify the figure.
Thank you, we corrected the figure.
- Throughout the manuscript, the authors have used both ‘comma’ and ‘period’ for the decimal point. Please change them all to periods.
Thank you, we corrected.